# A Systematic Review and Meta-Analysis of the Effectiveness of Non-Face-to-Face Coaching

**Yucheon Kim and Songyi Lee \*** 

Counselling and Coaching Department, Graduate School, Dongguk University-Seoul, 30, Pildong-ro 1 gil, Jung-gu, Seoul 04620, Republic of Korea
\* Correspondence: songyilee@empas.com or songyilee@dongguk.edu; Tel.: +82-10-6357-7310

**Abstract:** This study examined the effectiveness of non-face-to-face coaching in South Korea in order to present alternatives in the post-COVID-19 environment. The research collected domestic studies on non-face-to-face coaching in South Korea and analysed the studies through a systematic literature review and meta-analysis. Among 1081 papers retrieved from the database, we selected ten papers for meta-analysis. Using the random effect model to measure effect size, the total effect size of non-face-to-face coaching was 0.77. When we divided the effect of non-face-to-face coaching into psychological, cognitive, and physical effects, the cognitive effects were the largest. In addition, examining non-face-to-face coaching by type resulted in a larger effect size of web-based online coaching in comparison to telephone coaching. By contrast, the effect sizes of non-face-to-face coaching by subject had the largest effect size on subjects with the highest level of vulnerability. This study found that non-face-to-face coaching had a large effect, with relatively large cognitive and psychological effects. Future investigations should supplement the present research through follow-up studies on non-face-to-face coaching.

**Keywords:** non-face-to-face coaching; e-coaching; systematic literature review; meta-analysis



## 1. Introduction

Due to the COVID-19 pandemic, many face-to-face activities have been converted into non-face-to-face activities. Non-face-to-face is a term applied in various fields, including education and counselling [1], with a meaning of not being restrictive in multiple aspects, including the characteristic of 'not facing'. In addition to education and counselling, coaching has also been converted into a non-face-to-face activity. Coaching is a process that maximizes clients' potential so they can fulfil their goals. Although non-face-to-face coaching already occurred prior to the COVID-19 pandemic [2,3], interest in non-face-to-face coaching has been continuously increasing due to COVID-19.

Non-face-to-face coaching is similar to e-coaching: the coach does not meet clients in person but engages through distance, telephone, online, remote, virtual, and digital coaching. Non-face-to-face coaching also includes meeting clients through electronic means such as telephone, emails, and audiovisual or video meetings [3,4]. While non-face-to-face coaching has various terms, it refers to coaching clients using electronic media with no restrictions on time and place [5]. Thus, non-face-to-face coaching benefits clients because it has no limits on time and place [6]. Consequently, we see non-face-to-face coaching gradually replacing face-to-face coaching thanks to the internet [7].

In a study examining the effect of non-face-to-face coaching, Dwinger et al. [8] carried out telephone-based health coaching (TBHC). They found that non-face-to-face coaching helped patients to learn about their diseases and how to improve their condition. It also gave them the confidence to participate in treatment and to reach their targeted health goals. In addition, some studies indicate that e-coaching effectively reduces cardiovascular risks for ten years [9].

Another study [10] found digital text message coaching effective for personality change. In other research, Kettunen et al. [11] examined the effectiveness of digital coaching in promoting physical activities with the young elderly (65–75 years old). They learned that digital coaching positively affected self-efficacy, but stated that the study subjects recognized that it should be easy to use and attractive in order to fit the subjects.

Recently, as interest in non-face-to-face coaching has increased, scholars have conducted comprehensive studies on its effectiveness. For example, Akinosun et al. [12] stated that using digital technologies such as mobile phones, the internet, software applications, wearables, etc., is effective for patients with cardiovascular disease. Moreover, Gershkowitz et al. [13] indicated that digital health coaching's effects are similar to face-to-face or telephone coaching for preventing and maintaining long-term type 2 diabetes. However, Bevilacqua et al. [14] systematically analysed the effectiveness of health coaching for older people using digital technologies. They obtained a result indicating that coaching with technology integration rather than general treatment can benefit health. However, they could not get sufficient grounds for methods to maintain the advantage and permanence of the behavioural change due to the short-term intervention.

The research subjects In non-face-to-face coaching range from vulnerable people to those looking for prevention. For instance, Lambert and colleagues [15] researched the effectiveness of web-based stress management programs in vulnerable subjects, such as adults with cardiovascular disease. In addition, Thielecke et al. [16] have shown that telephone coaching prevents depression in farmers. Non-face-to-face coaching has also proven effective for traditional coaches and clients, and it is helpful in teachers' professional development relating to children's education [17].

While various studies on non-face-to-face coaching exist, scholars have conducted studies on non-face-to-face coaching in South Korea since 2008, despite the introduction of coaching having been more than 20 years ago [18,19]. For example, Kyoung Kim [20] developed a non-face-to-face coaching-based leadership program, which proved effective in career decision-making, self-efficacy, and resilience. Further, Choi and colleagues [21] showed the positive effect of non-face-to-face coaching on parenting. In addition, Lee and colleagues [22] identified six types through a study on college students' perceptions of non-face-to-face coaching.

Although there are some studies on non-face-to-face coaching in South Korea, there is no systematic literature review or meta-analysis [1]. Meta-analysis is a statistical method synthesizing a pooled estimate by combining estimates from two or more individual studies. In other words, it is a statistical technique used to evaluate effectiveness and efficiency by quantitatively calculating an integrated summary estimate of the results presented in studies [23].

Since non-face-to-face coaching is highly likely to be beneficial, it is important to examine its effectiveness in general and present ways for it to develop. Therefore, this study examines the effectiveness of non-face-to-face coaching and suggests how customers can continue to develop their lives. The study also demonstrates that non-face-to-face coaching can be a sustainable tool for individual development. We explore non-face-to-face coaching's effectiveness and how it can develop in a new environment that seeks more effective coaching. Hence, this study asks the following questions.

What is the size of the effect of non-face-to-face coaching?

What are the sizes of the effects of non-face-to-face coaching by type (cognitive, psychological, physical)?

What are the effect sizes of each type of non-face-to-face coaching (web-based online coaching, telephone coaching)?

What are the sizes of the effects of non-face-to-face coaching by subject and age?

What are the sizes of the effects of non-face-to-face coaching by operation (number of times, time)?

## 2. Method

### *2.1. Study Subjects*

This study collected and analysed research related to non-face-to-face coaching in South Korea. Achieving this study's purpose involved setting the population, intervention, comparison, outcomes, and study design (PICOS) in structured questions to clarify the critical concepts of the topic of the study [24].

### 2.1.1. Inclusion Criteria

The inclusion criteria for PICOS for research papers in this study included the following. The study subjects (P) were those receiving non-face-to-face coaching, and the intervention program (I) was non-face-to-face coaching. The control group included those not treated with non-face-to-face coaching. Thus, we could compare (C) the two populations. The effect of the relevant intervention (O) is the effectiveness of non-face-to-face coaching, and the study design (S) is the design of the pretest and posttest of the control group.

### 2.1.2. Exclusion Criteria

This research excluded low-quality study results from meta-analyses, and we conducted meta-analyses on papers published in journals listed in KCI. However, we included dissertations to supplement the representativeness of study results.

This study excluded the following: (1) papers and works in which the topic was not non-face-to-face coaching, (2) papers and works that did not contain the effect of non-face-to-face coaching, (3) papers that did not present sufficient data showing the effect of non-face-to-face coaching, and (4) qualitative study papers that did not show the quantitative effects of non-face-to-face coaching.

### 2.1.3. Search Strategy

This study followed the PRISMA guidelines for systematic literature review. PRISMA stands for "Reporting Items for Systematic Reviews and Meta-Analyses." We collected data through the following stages: literature verification, literature selection, selection criteria review, and final selection [25]. Regarding non-face-to-face coaching, we retrieved related data using the search engine RISS (Research Information Sharing Service) of the Korea Education & Research Information Service, the National Assembly Library of Korea, and KISS. Our target papers were those published from 2000 to 2021. Our search topics included: text coaching, telephone coaching, online coaching, digital coaching, web coaching, mobile coaching, artificial intelligence (AI) coaching, metaverse coaching, and non-face-to-face coaching. Our search retrieved 1081 papers—996 from RISS, 51 from the National Assembly Library, and 34 from KISS. Figure 1 illustrates the process of data review in our research. In this study, two researchers independently performed the procedure shown in Figure 1. In the event of objections in reviewing and selecting studies, the researchers resolved the objections through a continuous consultation and agreement process. Our systematic review protocol was registered in the International Prospective Register of Systematic Reviews (registration number: CRD42023429447).

Among the 1081 papers retrieved through the database, we excluded 782 papers, consisting of 304 papers, 453 books, and 25 research papers, by reviewing types and titles. Next, we excluded 270 topics unrelated to non-face-to-face coaching and five overlapping papers. Then, by re-reviewing the details of the papers, we excluded 14 with insufficient data and those containing qualitative research. Finally, we selected ten papers for the meta-analysis.

### 2.1.4. Data Extraction and Study Quality Assessment

We coded through the following procedure to collect papers related to non-face-to-face coaching and analyse the effect sizes. First, we recorded the paper's basic information (title, publication year, sample size), study design (independent and dependent variables,

measurement tools), and relevant variables. Then, a researcher with experience in meta-analysis studies did the coding. After completing the coding, the co-researcher reviewed the codes. After the review, the researchers discussed and revised the categories of individual variables. The process was continuous between the researchers in order to reach a final agreement.

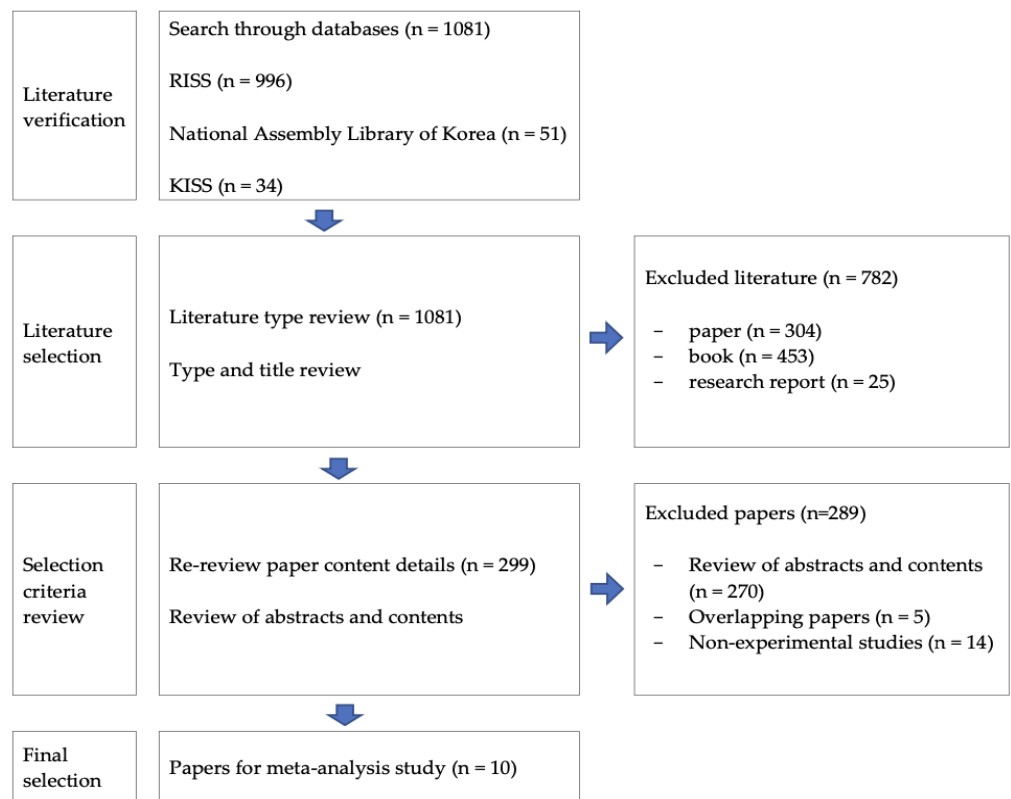

**Figure 1.** Process to select papers for research: PRISMA flow diagram.

We assessed the qualities of individual studies to examine the studies' validities regarding how well their results measured the actual effects and internal validity related to minimizing bias [26]. The assessment details concerning the quality of the target papers are as follows. We gave one point for each criterion if the study was affirmative to the following questions:

1. Was the sample frame appropriate to address the target population?
2. Were study participants recruited appropriately?
3. Was the sample size adequate?
4. Were the study subjects and setting described in detail?
5. Was data analysis conducted with sufficient coverage of the identified sample?
6. Were valid methods used for the identification of the condition?
7. Was the condition measured in a standard, reliable way for all participants?
8. Was there an appropriate statistical analysis?
9. Was the response rate adequate, and if not, was the low response rate managed appropriately?

We evaluated each target paper with the above qualitative assessment questions, and Supplementary Table S2 (Quality Assessment) contains the details. In addition, Table 1 presents the effect sizes, subject types, and coaching compositions of the studies included in this study according to the above criteria.

Table 1. Studies' details.

| No. | Researcher | Effect Size | Paper Type | Leading Variable | Dependent Variable | Subject | Age of Subject (Years) | Experimental Group | Control Group | Coaching Period | Number of Times in Coaching | Type |
|---|---|---|---|---|---|---|---|---|---|---|---|---|
| 1 | Cho, Y.Y. (2021) [27] | 0.896 | Dissertation | Online Coaching (Web-Based) | — Self-Management<br>— Motivation | College Student | 20 s | 7 | 9 | 7 | 7 | One-on-one online |
| 2 | Ryu, S.W. (2021) [28] | 0.769 | Journal Paper | Telephone Coaching | — Self-Management | Ordinary Person | 20 s–60 s | 20 | 20 | 8 | 8 | Utilizing a VR headset, smart band |
| 3 | H.Y. Kim et al. (2021) [29] | 0.803 | Journal Paper | Online Coaching (Web-Based) | — Self-Management<br>— Self-Efficacy<br>— Life Satisfaction<br>— Physical Health | Ordinary Person | 20 s~40 s | 8 | 9 | 8 | 8 | One-on-one online |
| 4 | A.K. Lee et al. (2021) [30] | 1.367 | Journal Paper | Online Coaching (Web-Based) | — Self-Efficacy<br>— Self-Management<br>— Life Satisfaction | Ordinary Person | 20 s~40 s | 10 | 6 | 8 | 8 | One-on-one online |
| 5 | Park, H.E. (2021) [31] | 1.545 | Journal Paper | Telephone Coaching | — Self-Management<br>— Psychological Health<br>— Self-Efficacy<br>— Physical Health | Type 2 Diabetic | 60 s | 28 | 28 | 12 | 2 | One-on-one phone call |
| 6 | Oh, E.Y. (2020) [32] | 1.367 | Journal Paper | Telephone Coaching | — Psychological Health<br>— Cognitive Function<br>— Executive Function | Patient with Ischemic Stroke | 60 s~70 s | 15 | 13 | 48 | 16 | One-on-one phone call |
| 7 | Lee, J.H. & Kim, S.H. (2016) [33] | 0.461 | Journal Paper | Telephone Coaching | — Physical Strength<br>— Physical Health | Elderly Woman | 60 s or older | 16 | 22 | 16 | 11 | One-on-one phone call |
| 8 | Hong, S.Y. (2016) [34] | 0.313 | Dissertation | Telephone Coaching | — Psychological Health<br>— Self-Efficacy<br>— Self-Management | Female | 30 s~40 s | 35 | 37 | 10 | 8 | One-on-one phone call |
| 9 | Kim, Y.J. & Lee, J.H. (2009) [35] | 0.644 | Journal Paper | Telephone Coaching | — Physical Strength<br>— Psychological Health | Physically Weak Elderly | 60 s or older | 27 | 26 | 10 | 8 | One-on-one phone call |
| 10 | S.H. Kim et al. (2008) [19] | 0.320 | Journal Paper | Telephone Coaching | — Physical Strength | Ordinary Person | 60 s or older | 16 | 27 | 16 | 10 | One-on-one phone call |

### 2.2. Effect Size Calculation

Effect size is a common unit for aggregating individual study results for integrated comparison. Commonly used effect sizes include standardised mean value, correlation coefficient I, odds ratio, etc. In this study, we compared and analysed effect sizes using the standardised mean change difference between the group that received non-face-to-face coaching and those that did not receive non-face-to-face coaching. The standardised mean difference is the standardised mean difference between the posttest and pretest in the experimental and control groups [36]. We interpreted the meaning of the study results based on the effect sizes. An effect size smaller than 0.2 is a 'small effect,' an effect size around 0.5 is a 'medium effect,' and an effect size of 0.8 or greater is a 'large effect' [37]. In this study, we analysed the effect sizes using the CMA (Comprehensive Meta-Analysis) 3.0 program.

### 2.3. Publication Bias Verification

We verified publication bias because study results that are positive and statistically significant are more likely to be published than those that are not positive and statistically significant [38]. In the absence of publication bias, studies will symmetrically distribute around the aggregated effect size, and in the presence of publication bias, studies will concentrate on the other side.

In Figure 2, the *x*-axis represents the effect size, and the *y*-axis represents the number of samples. Examining the funnel plot in Figure 2 for publication bias, we find that the left and right sides are not approximately symmetrical.

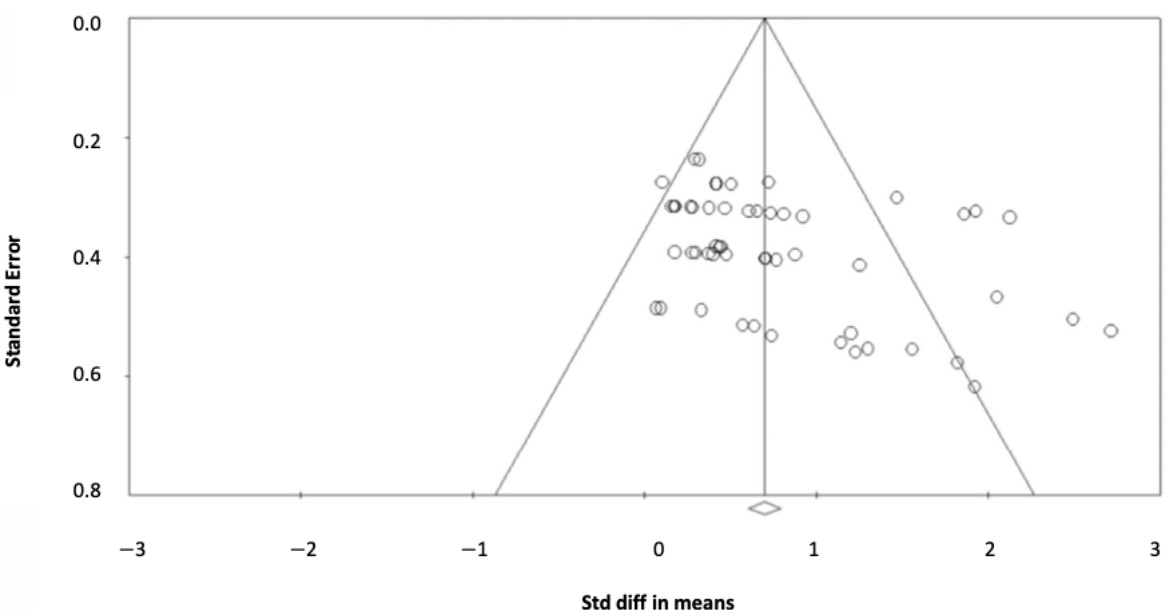

**Figure 2.** Funnel plot of standard error by Fisher's Z.

To verify publication bias, we estimated the influence of the excluded studies using the trim and fill method [39], as shown in Table 2. According to the result, the influence was not large enough to hinder the validity of the estimation of the effect of coaching. The funnel plot is not a proof of error but a tool for raising the possibility of bias [40]. Therefore, these results indicate that the publication bias is not severe.

### 2.4. Analysis Model and Homogeneity Test

In a meta-analysis, computational models for calculating the average effect size include fixed and random effect models [30]. The fixed-effect model estimates the effect size of the same population, whereas random-effect models estimate the average effect size of distributions of different populations.

**Table 2.** Duval and Tweedie's trim and fill.

| | | Fixed Effects | | | Random Effects | | | Q Value |
|---|---|---|---|---|---|---|---|---|
| | Studies Trimmed | Point Estimate | Lower Limit | Upper Limit | Point Estimate | Lower Limit | Upper Limit | |
| Observed Values | | 0.69866 | 0.60367 | 0.79365 | 0.77520 | 0.61123 | 0.93918 | 151.68911 |
| Adjusted Values | 3 | 0.73887 | 0.64621 | 0.83153 | 0.81449 | 0.65167 | 0.97731 | 165.89359 |

We conducted a homogeneity test to determine whether the results of individual studies were from the same population. Homogeneity tests evaluate whether the effect sizes extracted from individual studies are from the same population. We can use a fixed-effect model if individual studies are homogeneous, i.e., extracted from the same population through a homogeneity test. On the other hand, if a homogeneity test produces a statistically significant result, we would use a random-effect model to measure effect size, assuming that individual study results came from different populations.

The null hypothesis in a homogeneity test examines whether the effect sizes are homogeneous and if they are homogeneous, then we can regard them as extracted from the same population. In Table 3, Q-value means the observed variance of each effect size, and df (degrees of freedom) is the value of the expected variance when assuming all the effect sizes of the populations of individual studies are the same. If the value obtained by subtracting the expected variance (df) from the total observed variance (Q-value)—that is, the degree of variance based on the actual effect difference between studies—is Q > df, then the effect sizes of the populations of individual studies are different from each other.

**Table 3.** Homogeneity test.

| Model | Q-Value | *df* (Q) | *p*-Value | I-Squared |
|---|---|---|---|---|
| Fixed | 151.698 | 54 | 0.000 | 64.401 |

In this study, since the Q-value, 151.698, is larger than df 54, the effect sizes of populations are different. In addition, since the significance probability (*p*-value) of the homogeneity test was smaller than 0.10, the heterogeneity of the effect sizes is significant. Therefore, using the random-effect model is appropriate. When the I-squared value representing the ratio of actual variance to total variance is 25%, the heterogeneity is small; when the I-squared value is 50%, the heterogeneity is medium; when the value is 75%, the heterogeneity is very large. This study's I-squared value was 64.401, indicating heterogeneous effect sizes.

Since all of the studies' subjects, intervention methods, periods, etc., are different, the effect sizes of the studies' populations also differ. Therefore, we used the random-effect model to calculate the average effect size in this study.

*2.5. Effect Sizes by Research Type*

The effect sizes by research paper type are in Table 4. The effect size of journal papers (d = 0.809) was larger than that of dissertations (d = 0.670).

**Table 4.** Effect sizes by paper type.

| Division | *k* | *d* | SE | 95% CI | |
|---|---|---|---|---|---|
| | | | | LL | UL |
| Journal Paper | 48 | 0.809 | 0.093 | 0.626 | 0.992 |
| Dissertation | 7 | 0.670 | 0.074 | 0.525 | 0.816 |

Notes: *k*, number of samples; *d*, Cohen's d effect size; SE, standard error; 95% CI; 95% confidence interval; LL, lower limit; UL, upper limit.

### 3. Results

*3.1. Overall Effect Size*

In this study, since the Q-value, 151.698, is larger than df 54, the effect sizes of populations are different. In addition, the I-squared value was 64.401, indicating heterogeneous effect sizes, as following Table 4. We used the random effect model to verify the overall effect size. Table 5 shows that the effect size for 55 effect size cases was 0.775, and the 95% confidence interval for the overall effect was 0.611–0.939. Although the effect size is slightly smaller than 0.8, Cohen's [28] standard indicates that this result is significant for the non-face-to-face coaching effect.

**Table 5.** Overall effect size, random effects analysis.

| k | ES | 95% CI | | Q-Value | df | *p*-Value | I² |
|---|---|---|---|---|---|---|---|
| | | **Lower** | **Upper** | | | | |
| 55 | 0.775 | 0.6111 | 0.939 | 151.639 | 54 | 0.000 | 64.401 |

Note: *k*, number of cases of effect sizes; ES, effect size; Q, observed variance of each effect size; df, degree of freedom; I², heterogeneity (ratio of actual variance to total variance).

*3.2. Non-Face-to-Face Coaching Effect Size*

We divided the effect sizes by treatment effect of the non-face-to-face coaching effects into psychological, cognitive, and physical effects; Table 6 contains the details. Among the non-face-to-face coaching effects, cognitive effects (d = 1.494) were the largest, followed by psychological effects (d = 0.943), and physical effects (d = 0.428). In the three categories of effect sizes, the largest effect sizes were in psychological health (d = 128), cognitive function (d = 1.792), and physical health (d = 0.551).

**Table 6.** Non-face-to-face coaching effect size.

| Division | k | d | SE | 95% CI | |
|---|---|---|---|---|---|
| | | | | **LL** | **UL** |
| Psychological Effect | 24 | 0.943 | 0.133 | 0.682 | 1.205 |
| — Psychological Health | 4 | 1.128 | 0.456 | 0.234 | 2.022 |
| — Motivation | 2 | 0.606 | 0.364 | −0.108 | 1.320 |
| — Life Satisfaction | 2 | 0.615 | 0.579 | −0.519 | 1.749 |
| — Self-Management | 12 | 0.841 | 0.131 | 0.585 | 1.097 |
| — Self-Efficacy | 4 | 1.089 | 0.497 | 0.115 | 2.063 |
| Cognitive Effect | 8 | 1.494 | 0.315 | 0.291 | 0.565 |
| — Cognitive Function | 6 | 1.792 | 0.366 | 1.076 | 2.509 |
| — Executive Function | 2 | 0.656 | 0.276 | 0.116 | 1.197 |
| Physical Effect | 23 | 0.428 | 0.070 | 0.291 | 0.565 |
| — Physical Health | 6 | 0.551 | 0.237 | 0.086 | 1.016 |
| — Physical Strength | 17 | 0.377 | 0.078 | 0.223 | 0.531 |

Notes: *k*, number of samples; *d*, Cohen's d effect size; SE, standard error; 95% CI; 95% confidence interval; LL, lower limit; UL, upper limit.

*3.3. Effect Sizes by Coaching Type*

The effect sizes of non-face-to-face coaching by coaching type are in Table 7. The effect size of online coaching using Zoom (d = 0.914) in non-face-to-face coaching was larger than that of telephone coaching (d = 0.744).

**Table 7.** Effect size by type of coaching.

| Division | $k$ | $d$ | SE | 95% CI | |
| --- | --- | --- | --- | --- | --- |
| | | | | LL | UL |
| Online (Web-Based) Coaching | 13 | 0.914 | 0.170 | 0.581 | 1.247 |
| Telephone Coaching | 42 | 0.744 | 0.094 | 0.559 | 0.929 |

Notes: $k$, number of samples; $d$, Cohen's d effect size; SE, standard error; 95% CI; 95% confidence interval; LL, lower limit; UL, upper limit.

### 3.4. Effect Size by Coachee

The effect sizes of non-face-to-face coaching by coachee are in Table 8. Among the non-face-to-face coaching effects, those with a high level of vulnerability (d = 1.041) were the largest, followed by those with a low level of vulnerability (d = 0.586), and a medium level of vulnerability (d = 0.389). Among the effects on persons with a high level of vulnerability, those with diabetes (d = 1.481) were the largest, followed by older females (d = 0.461), with the largest effect size in the medium vulnerability group. Among those with a low level of vulnerability, college students (d = 0.896) showed the largest effect size.

**Table 8.** Effect sizes by the level of vulnerability coachees.

| Division | $k$ | $d$ | SE | 95% CI | |
| --- | --- | --- | --- | --- | --- |
| | | | | LL | UL |
| High Vulnerability | 18 | 1.041 | 0.081 | 0.882 | 1.199 |
|  − Persons with Diabetes | 4 | 1.481 | 0.153 | 1.181 | 1.782 |
|  − Persons with Ischemic Stroke | 9 | 1.221 | 0.142 | 0.942 | 1.501 |
|  − Physically Weak Elderly | 5 | 0.588 | 0.128 | 0.337 | 0.838 |
| Medium Vulnerability | 12 | 0.389 | 0.095 | 0.203 | 0.576 |
|  − Child Patient's Mother | 3 | 0.313 | 0.137 | 0.044 | 0.581 |
|  − Female Elderly | 9 | 0.461 | 0.133 | 0.201 | 0.721 |
| Low Vulnerability | 25 | 0.586 | 0.078 | 0.432 | 0.740 |
|  − Ordinary Person | 21 | 0.557 | 0.082 | 0.396 | 0.718 |
|  − College Student | 4 | 0.896 | 0.266 | 0.375 | 1.4162 |

Notes: $k$, number of samples; $d$, Cohen's d effect size; SE, standard error; 95% CI; 95% confidence interval; LL, lower limit; UL, upper limit.

### 3.5. Effect Sizes by Coaching Period and Time

The sizes of the effects of non-face-to-face coaching by period and time are in Table 9. The effect sizes by coaching period were the largest when the coaching period was 12 weeks (d = 1.545). The effect sizes by the number of coaching sessions were the largest when coaching was 0.160 times per week (d = 1.545). The effect sizes by the total number of coaching sessions were the largest when the total number of sessions was two (d = 1.545). While this effect size was the largest, Park [31] conducted two sessions after the client had already received two prior coaching sessions. The effect sizes by the time spent per coaching session were the largest when the time spent per coaching session was 80 min (d = 0.896).

**Table 9.** Effect size by coaching period and time.

| Division | k | d | SE | 95% CI | |
|---|---|---|---|---|---|
| | | | | LL | UL |
| Coaching Period (weeks) | | | | | |
| − Seven Weeks | 4 | 0.896 | 0.226 | 0375 | 1.416 |
| − Eight Weeks | 13 | 0.843 | 0.139 | 0.571 | 1.115 |
| − Ten Weeks | 8 | 0.502 | 0.163 | 0.183 | 0.822 |
| − 12 Weeks | 4 | 1.545 | 0.322 | 0.913 | 2.178 |
| − 16 Weeks | 17 | 0.379 | 0.086 | 0.211 | 0.547 |
| − 48 Weeks | 9 | 1.367 | 0.296 | 0.786 | 1.949 |
| Number of Times Per Week | | | | | |
| − 0.160 Times | 4 | 1.545 | 0.322 | 0.913 | 2.178 |
| − 0.330 Times | 9 | 1.367 | 0.296 | 0.786 | 1.949 |
| − 0.620 Times | 8 | 0.320 | 0.112 | 0.100 | 0.541 |
| − 0.680 Times | 9 | 0.461 | 0.133 | 0.201 | 0.721 |
| − 0.800 Times | 8 | 0.502 | 0.163 | 0.183 | 0.822 |
| − 1.000 Times | 17 | 0.842 | 0.112 | 0.622 | 1.062 |
| Total Number of Times | | | | | |
| − Two Times | 4 | 1.545 | 0.322 | 0.913 | 2.178 |
| − Seven Times | 4 | 0.896 | 0.266 | 0.375 | 1.416 |
| − Eight Times | 8 | 0.320 | 0.112 | 0.100 | 0.541 |
| − Ten Times | 9 | 0.461 | 0.133 | 0.201 | 0.721 |
| − 11 Times | 9 | 0.461 | 0.133 | 0.201 | 0.721 |
| − 16 Times | 9 | 1.367 | 0.296 | 0.786 | 1.949 |
| Time Per Session | | | | | |
| − 15 min | 5 | 0.644 | 0.279 | 0.098 | 1.190 |
| − 20 min | 9 | 0.461 | 0.133 | 0.201 | 0.721 |
| − 30 min | 24 | 0.879 | 0.147 | 0.590 | 1.167 |
| − 60 min | 13 | 0.843 | 0.139 | 0.571 | 1.115 |
| − 80 min | 4 | 0.896 | 0.266 | 0.375 | 1.416 |

Notes: *k*, number of samples; *d*, Cohen's d effect size; SE, standard error; 95% CI; 95% confidence interval; LL, lower limit; UL, upper limit.

## 4. Discussion

This study analysed the effect sizes of non-face-to-face coaching through meta-analysis. It aggregated the results of previous studies on non-face-to-face coaching, examining effect size correlations to compare individual effects. The analysis results of ten papers related to non-face-to-face coaching are as follows.

First, we measured effect sizes with a random effect model to obtain the effect sizes in the meta-analysis, resulting in 0.77 as the overall effect size of non-face-to-face coaching. Concerning the effect size, according to Cohen [28], researchers can interpret the effect size to be close to a large effect size, making it possible to suggest that non-face-to-face coaching is effective. This study examined papers on the effects of non-face-to-face coaching in South Korea, finding results similar to those of various overseas studies showing the effectiveness of non-face-to-face coaching [9,41].

However, some overseas studies have suggested that non-face-to-face coaching cannot obtain sufficient effects alone. For example, in a study by Jones et al. [42], some participants said phone and internet-based coaching had advantages over face-to-face coaching. However, they suggested that the prospect of a mixture of non-face-to-face and face-to-face coaching using technologies was bright. In addition, a systematic review by van Veen [43] indicated that basic e-coaching was ineffective for patient rehabilitation.

The methods of non-face-to-face coaching in studies in South Korea, the subjects of the current analysis, were as simple as online and telephone coaching. There were no mixed studies and insufficient studies to compare non-face-to-face and face-to-face coaching. Therefore, experimental studies on more diverse non-face-to-face coaching methods are necessary to vitalize non-face-to-face coaching in South Korea. In particular, Zhang et al. [44] studied speech effects in the metaverse. Thus, we need research to apply this metaverse to coaching and develop non-face-to-face coaching.

Second, cognitive effects were the largest when we examined the psychological, cognitive, and physical effects of non-face-to-face coaching. For instance, Schouten et al. [45] found meaningful results supporting cognitive learning through digital coaching for low-literate individuals. In addition, psychological effects were significant, but physical effects were not large when compared to other areas. Therefore, face-to-face coaching could help increase physical effects. Furthermore, coaches should determine the coaching methods based on the coaching purpose and include the complementary activity of face-to-face coaching according to the client's needs. This approach will vitalize more effective non-face-to-face coaching.

Third, when we examined the effect sizes of non-face-to-face coaching by type, we found the effect size of web-based online coaching was larger than telephone coaching. In telephone coaching, among the types of non-face-to-face coaching, communications can be limited because the other party cannot see non-verbal elements, such as facial expressions and gestures. Thus, web-based online coaching is more effective for communication because the parties can see each other's facial expressions and gestures [46].

In addition, based on our analysis, although the division of ages is unclear, except in the study conducted by Hong [34], studies involving web-based online coaching were only in studies with subjects in their 20 s–40 s. This effect of online coaching in subjects in relatively young age groups indicates their capability to handle online technology. Therefore, although media can enable smooth communication with clients in non-face-to-face coaching, selecting the medium requires consideration of the ages of clients.

Fourth, the effects of non-face-to-face coaching by coachee were largest in persons with a high level of vulnerability. This study's subjects with a high level of vulnerability included persons with diabetes, ischemic stroke patients, and the physically weak elderly, and all received telephone coaching. Therefore, non-face-to-face coaching using telephones is effective for persons with a high level of vulnerability. In addition, Kettunen et al. [11] showed that digital coaching motivates young older people towards physical activities, which aligns with our study's results.

Yousuf [9] found severe limitations when e-coaching older people using web-based tools. Moreover, Mahdaria and Restuaji [47] showed that the effectiveness of online coaching depends on the reliability of the internet connection, suggesting a need for various platforms that can adapt to technical problems. In addition, while analysing two papers in a systematic literature review of digital health coaching programs for retired seniors to become re-employed in the community, Stara et al. [48] proposed a user-centred design approach for older adults. The studies show that for non-face-to-face coaching to be more suitable for the subject, the subject must be able to use the method easily. Therefore, to vitalize non-face-to-face coaching in South Korea, coaches should prepare various platforms according to the subject's ability to handle individual skills.

Fifth, in this study, the effect sizes of non-face-to-face coaching by coaching period were the largest when the coaching period was 12 weeks, followed by a coaching period of 48 weeks. However, the effect size when the coaching period was 16 weeks was small.

In studies implementing a coaching period of 16 weeks [33,35], non-face-to-face coaching lasted 15 to 20 min per session. This length is characteristic of other studies in which coaching lasted 30 to 60 min. In addition, the total effect size was largest when there were two coaching sessions, followed by 16 sessions. However, it is difficult to comprehend two as the most effective number of sessions because Park [31] first conducted face-to-face coaching and then non-face-to-face coaching, but studied only the effectiveness of non-face-to-face coaching. Therefore, based on the results of this study, the effect size increases to some extent as the number of coaching sessions increases.

Our results align with those of Theeboom et al. [49], indicating that as the coaching period and the number of coaching sessions increase, we see larger effects in the results at the individual level, such as goal achievement, self-efficacy, and quality of life. Regarding the coaching period, a study by Grant [50] compared participants who received coaching for 16 weeks with those who received coaching for 32 weeks. The study's results indicated that participants coaching for 32 weeks saw higher effects on goal achievement and mental health. Therefore, coaching effects increase along with the coaching period.

On the other hand, when we examined the coaching period in our study, there was no significant difference in the effect between 12 weeks and 48 weeks, but the effect for 48 weeks was smaller than for 12 weeks. Therefore, although the coaching period should be sufficiently continuous and long, there might be a reduced effect in cases where the coaching period is excessively long. The preceding shows the importance of conducting non-face-to-face coaching appropriately and consistently. In cases where a client wishes to achieve goals through coaching, coaches should adequately adjust the period and number of non-face-to-face coaching sessions to obtain effective results.

## 5. Limitation

Despite the recent application of various non-face-to-face coaching methods, the present study reviewed research on non-face-to-face coaching in South Korea. However, since studies related to non-face-to-face coaching in South Korea are insufficient, there may be limitations in generalizing the results of the meta-analysis of studies to non-face-to-face coaching. Therefore, future investigations should supplement the present research through follow-up studies as more studies on non-face-to-face coaching accumulate. In addition, this study's qualitative analysis of the literature was narrow. Hence, future research should conduct a qualitative analysis of the study subjects. Finally, regarding the effects of non-face-to-face coaching, future research should add a meta-analysis of qualitative studies with only qualitative research papers, thereby adding various meta-studies related to non-face-to-face coaching.

## 6. Conclusions

This study obtained the following conclusions through meta-analyses of the effects of non-face-to-face coaching. First, even though the meta-analyses showed that the effect size of non-face-to-face coaching was large, various experimental studies on additional non-face-to-face coaching are needed. Second, the meta-analyses showed that in non-face-to-face coaching, cognitive and psychological effects were relatively higher than physical effects. Third, the effects were different depending on the types of non-face-to-face coaching. Web-based online coaching showed a higher effect size than telephone coaching. Fourth, the meta-analyses showed that the effect size of non-face-to-face coaching was relatively large for subjects with a high level of vulnerability. Fifth, the meta-analyses showed that the period and number of sessions of non-face-to-face coaching affected the size of the effect of coaching.

We expect the results of this study to provide meaningful data for the vitalization of non-face-to-face coaching in South Korea. Coaching relies heavily on various forms of communication and technology, and coaching places and forms can apply coaching methods differently from traditional ones, thus, coaching skills and methods are innovative [51]. Furthermore, as non-face-to-face activities inevitably increased due to COVID-19,

coaching is also progressing considerably by using the non-face-to-face method. Therefore, researchers should continuously study and develop coaching to pursue innovation according to change, while remaining faithful to the basics, to solve the complexity of individuals and relationships.

**Supplementary Materials:** The following supporting information can be downloaded at: https://www.mdpi.com/article/10.3390/su15129727/s1, Table S1: PRISMA Checklist, Table S2: Quality assessment.

**Author Contributions:** Data curation, Y.K.; formal analysis, S.L.; methodology, Y.K.; validation, S.L.; writing—original draft, Y.K. and S.L.; writing—review & editing, Y.K. and S.L. All authors have read and agreed to the published version of the manuscript.

**Funding:** This research received no external funding.

**Institutional Review Board Statement:** Not applicable.

**Informed Consent Statement:** Not applicable.

**Data Availability Statement:** Data from the study are available upon request.

**Conflicts of Interest:** The authors declare no conflict of interest.

**Pre-Registration:** We prospectively registered a protocol of our meta-analysis in the International Prospective Register of Systematic Reviews CRD42023429447 on 5 June 2023.

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
