# Peer review of "A Systematic Review and Meta-Analysis of the Effectiveness of Non-Face-to-Face Coaching"

_sustainability, doi:10.3390/su15129727_

Round 1
Reviewer 1 Report
This study examined the effectiveness of non-face-to-face coaching in South Korea to present alternatives in the post-COVID-19 environment. The study collected domestic studies on non-face-to-face coaching in South Korea and analysed the studies through a systematic literature review and meta-analysis. Among 1,081 papers retrieved from the database, they selected ten papers for meta-analysis study. I suggested to add some related works about metaverse such as "Xulong Zhang, Jianzong Wang, Ning Cheng, Jing Xiao (2022). MetaSpeech: Speech Effects Switch Along with Environment for Metaverse".
Using the random effect model to measure effect size, the total effect size of non-face-to-face coaching was 0.77. When divided the effect of non-face-to-face coaching into psychological, cognitive, and physical effects, the cognitive effects were the largest.
In addition, examining non-face-to-face coaching by type resulted in a larger effect size of web-based online coaching than telephone coaching. In contrast, the effect sizes of non-face-to-face coaching by subject found the largest effect size on subjects with the highest level of vulnerability.
This study found a large effect of non-face-to-face coaching, with relatively large cognitive and psychological effects on non-face-to-face coaching.
Author Response
Thank you for your feedback. We added more references as advised.
Reviewer 2 Report
Dear Authors
The paper is right written with comprehensive text, but I believe that the paper doesn't add knowledge to the research field, is not original, with a plenty of open question not answered. In my opinion, the manuscript should be rejected.
My arguments are the followings:
1.- the authors don’t really do a meta-analysis, the manuscript is systematic review of literature
2.- the abstract is incomprehensible
3.- the study is very limited with only 10 paper reviewed, that is a 1% of the whole example.
4.- Some analysis incomprehensible
5.- the research questions in my opinion are not answered.
6.- Studies details are not used in the research (table 1)
7.- figure 2 is not symmetrical
8.- Many of the tables, data and numbers in the manuscript are not known for what they are used.
9.- the sections of results and conclusions are very poor, not well explained and not clear. It is almost impossible obtain the conclusions from the results of the research.
Author Response
1.- the authors don’t really do a meta-analysis, the manuscript is systematic review of literature
Authors’ response: Thank you for your feedback. However, a meta-analysis research involves the use of a statistical method to evaluate effectiveness and efficiency by quantitatively calculating an integrated summary estimate of the results presented in studies (Kang, H., 2015). Our research utilized the CMA (Comprehensive Meta-Analysis) 3.0 program to derive our statistical data that we used to conduct our meta-analysis, where we did a comprehensive review of the effect size of the ten literature papers based on variables. Therefore, this study would be considered a meta-analysis research. We have added more details for clarification.
2.- the abstract is incomprehensible
Authors’ response: We have revised for clarification.
3.- the study is very limited with only 10 paper reviewed, that is a 1% of the whole example.
Authors’ response: We had discussed this in the limitation section. However, meta-analysis studies can be done with a small number of studies, and we have added a reference for this in the introduction section as follows. Meta-analysis refers to a statistical method that synthesizes a pooled estimate by combining estimates from two or more individual studies (Kang, H., 2015).
4.- Some analysis incomprehensible
Authors’ response: Thank you for your feedback. We were unsure which specific areas you were referring to. So, we did a thorough review and did our best to revise and clarify as much as needed.
5.- the research questions in my opinion are not answered.
Authors’ response: We think the ordering of the questions and how they are answered in the results section may have caused your confusion. So, we have revised this for clarification.
6.- Studies details are not used in the research (table 1)
Authors’ response: Thank you for your feedback. We are not sure exactly what you meant by this feedback. We used all the studies details for the results and discussion section. Actually, we noticed that we were missing the paper type, so we added that to Table 1.
7.- figure 2 is not symmetrical
Authors’ response: Thank you for your feedback. We have added Table2 (Duval and Tweedie’s trim and fill) and revised for clarification. Also, “The funnel plot is not a proof of error but a tool for raising the possibility of bias” (Hwang, 2019). Our results indicated that publication bias is not severe and appropriate for analysis. We have added this reference for a better understanding of the funnel plot.
8.- Many of the tables, data and numbers in the manuscript are not known for what they are used.
Authors’ response: Thank you for your feedback. We are not sure exactly what you meant by this feedback. We did provide an explanation for all the tables and data in results section, and discussed about the data we have found in our discussion section. We did the best we could to review and revise any areas that looked confusing.
9.- the sections of results and conclusions are very poor, not well explained and not clear. It is almost impossible obtain the conclusions from the results of the research.
Authors’ response: Thank you your feedback. However, based on your specific feedbacks, we felt that there were misunderstandings and confusions that arose, which we did our best to clarify and address. We have also thoroughly reviewed and revised certain areas of our results and conclusions section to make sure we can clarify and respond to the concerns mentioned.
Reviewer 3 Report
The article addresses a topic of interest to the scientific community. This study examined the effectiveness of non-face-to-face coaching in South Korea, and analysed the studies through a systematic literature.
This study followed the PRISMA guidelines for systematic literature review. The methodology and procedure followed is well described. The conclusions are clear. In addition, the limitations of the study are presented, which may contribute to the definition of future lines of action.
I recommend expanding the theoretical basis in the Introduction section. In its present state it is reduced.
Author Response
Thank you for your feedback. We have revised accordingly.
Reviewer 4 Report
this is an interesting study. The overall design was good. The writing was clear. The biggest problem has to do with the sample size and the publication bias.
Author Response
Thank you for your feedback. In terms of the sample size, we had discussed this in the limitation section. However, meta-analysis studies can be done with a small number of studies, and we have added a reference for this in the introduction section as follows. Meta-analysis refers to a statistical method that synthesizes a pooled estimate by combining estimates from two or more individual studies (Kang, H., 2015). In terms of the publication bias, we have added Table 2 (Duval and Tweedie’s trim and fill) and revised for clarification. Our results indicated that publication bias is not severe and appropriate for analysis. We have added this reference for a better understanding of the funnel plot.
Round 2
Reviewer 2 Report
Dear Authors,
thank you for taking into account all my comments and suggestions , and for improving the paper.
My decision is accept in present form